# *Culex pipiens* crossing type diversity is governed by an amplified and polymorphic operon of *Wolbachia*

Manon Bonneau[1], Celestine Atyame[1,2], Marwa Beji[3], Fabienne Justy[1], Martin Cohen-Gonsaud[4], Mathieu Sicard[1] & Mylène Weill[1]

*Culex pipiens* mosquitoes are infected with *Wolbachia* (*w*Pip) that cause an important diversity of cytoplasmic incompatibilities (CIs). Functional transgenic studies have implicated the *cidA-cidB* operon from *w*Pip and its homolog in *w*Mel in CI between infected *Drosophila* males and uninfected females. However, the genetic basis of the CI diversity induced by different *Wolbachia* strains was unknown. We show here that the remarkable diversity of CI in the *C. pipiens* complex is due to the presence, in all tested *w*Pip genomes, of several copies of the *cidA-cidB* operon, which undergoes diversification through recombination events. In 183 isofemale lines of *C. pipiens* collected worldwide, specific variations of the *cidA-cidB* gene repertoires are found to match crossing types. The diversification of *cidA-cidB* is consistent with the hypothesis of a toxin–antitoxin system in which the gene *cidB* co-diversifies with the gene *cidA*, particularly in putative domains of reciprocal interactions.

[1] Institut des Sciences de l'Evolution de Montpellier (ISEM), UMR CNRS-IRD-EPHE-Université de Montpellier, Place Eugène Bataillon, 34095 Montpellier, France. [2] Processus Infectieux en Milieu Insulaire Tropical (PIMIT), UMR CNRS-INSERM-IRD-Université de La Réunion, Sainte-Clotilde, Ile de La Réunion, 97490, France. [3] Institut Pasteur Tunis, Laboratory of Epidemiology and Veterinary Microbiology, University of Tunis El Manar, 1068 Tunis, Tunisia. [4] Centre de Biochimie Structurale (CBS), UMR CNRS-INSERM-Université de Montpellier, 29 rue de Navacelles, 34090 Montpellier, France. Mathieu Sicard and Mylène Weill contributed equally to this work. Correspondence and requests for materials should be addressed to M.S. (email: mathieu.sicard@umontpellier.fr) or to M.W. (email: mylene.weill@umontpellier.fr)

The most common way by which *Wolbachia* bacteria spread within insect populations is cytoplasmic incompatibility (CI), which results in the early death of embryos[1] when males infected with a given *Wolbachia* strain mate with uninfected females or females infected with an incompatible strain. The most popular model to date for conceptualizing CI involves the secretion, by *Wolbachia*, of a "modification" factor (mod, or toxin) in the sperm that impairs early embryogenesis, and a *Wolbachia* "rescue" factor (resc, or antitoxin) produced in the oocyte, which allow for a viable, diploid zygote to develop if the cross is compatible[2–4]. Previous studies searching for *Wolbachia* effectors involved in CI by genomic comparison between mutualistic and manipulative *Wolbachia* revealed a higher number of genes with ankyrin repeats in the latter[5, 6]. Ankyrin repeat genes were thus considered as potential mod and resc candidates due to their role in protein–protein interactions and cellular cycle regulation[7]. However, to date no correlation between the distributions of any ankyrin repeat genes and CI patterns was demonstrated[8]. The first convincing candidate gene for involvement in CI encodes a protein called CidA (WP0282), secreted by *Wolbachia* into *Culex pipiens* sperm[9]. The *cidA* gene is part of an operon also containing a second gene, *cidB* (WP0283). CidA and CidB proteins directly interact with each other as demonstrated by pull-down experiments[10]. The *cidA* and *cidB* genes (named *cifA* and *cifB* in *w*Mel[11]) are associated with the prophage WO modules[11], and paralogs of these genes have been found in all published genomes from *Wolbachia* strains known to induce CI in insects[9, 11, 12]. The co-expression of *cidA*/*cidB* (or *cifA*/*cifB*) transgenes in *Drosophila melanogaster* reproduces the disturbance of the first embryonic division hallmark of CI, leading to embryo death[10, 11]. In this context, disruption of the deubiquitylating (DUB) domain of CidB restores normal embryogenesis, demonstrating the contribution of this domain to the "mod" factor function[10]. The role of CidA in the mod-resc CI system is much more debated. Data obtained on yeast suggest that CidA may rescue CidB[10], but CifA has also been described as a CifB elicitor in *D. melanogaster*[11].

*cidA*-*cidB* operon is involved in the CI induced by infected males in the progenies of uninfected females[10, 11], but the genetic basis of CI diversity (i.e., compatibility, uni or bidirectional incompatibility) between hosts infected with different *Wolbachia* strains has yet to be investigated. We addressed this question in *C. pipiens*, a powerful model in which hundreds of crosses between lines sampled worldwide have revealed unprecedented CI diversity[13–15]. This unique CI diversity is solely governed by *w*Pip diversification in this species complex since no other manipulative endosymbionts are present and no host genetic background influence has been yet demonstrated[13, 15]. All *C. pipiens* individuals are infected with *Wolbachia*, which recently diverged into five distinct phylogenetic groups (*w*PipI to *w*PipV)[16]. The *w*Pip strains of the same group generally ensure compatibility between their hosts, whereas intergroup crosses are more likely to be incompatible[13]. An analysis of multiple crosses concluded that each *Wolbachia* (*w*Pip) genome must contain several mod and resc factors to account for the high diversity of CI in *C. pipiens*[15, 17]. These multiple mod and resc factors could theoretically be encoded by different copies (i.e., variants) of same mod/resc genes or different mod and resc genes within the same *w*Pip genome[15, 17]. It therefore seems likely that (i) different copies (i.e., variants) of *cidA*-*cidB*, (ii) paralogs of these genes resulting from ancient duplication events[10, 11, 17], or (iii) totally unrelated genes[13], could be involved in CI diversity in *C. pipiens*.

We thus investigated the contribution of polymorphism of the *cidA*-*cidB* operon and its *cinA*-*cinB* paralog with predicted nuclease activity[10] to CI diversity in *C. pipiens* with the major hypothesis that if these genes encode for CI diversity determinants, they should be different between *C. pipiens* lines with different crossing types. We show that all tested *w*Pip genomes contain several variants of the *cidA*-*cidB* operon while it is not the case for *cinA*-*cinB* operon. The *cidA*-*cidB* repertoires of variants differ between *w*Pip strains from different groups exhibiting different crossing types. In 180 isofemale lines of *C. pipiens* exclusively infected with *w*PipIV strains, specific variations of the *cidA*-*cidB* gene repertoires are found to match crossing type

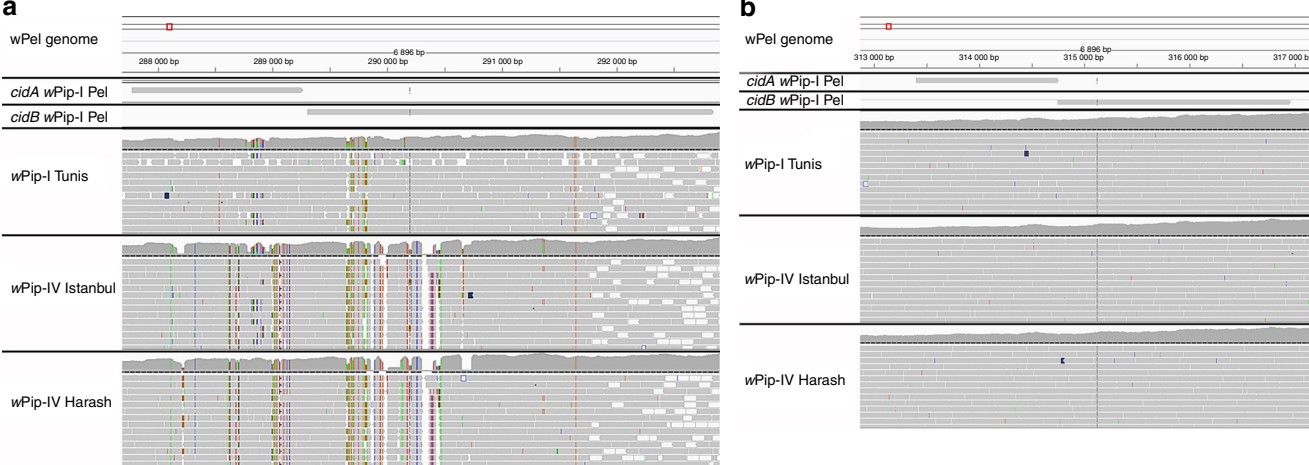

**Fig. 1** The *cidA*/*cidB* operon displays polymorphism within and between different *w*Pip strains, whereas no polymorphism is observed for the *cinA*/*cinB* operon. Mapping onto the reference genome *w*Pip_Pel of the Illumina reads from the Tunis line infected with *w*PipI, and from the Harash and Istanbul lines infected with *w*PipIV. Colored residues are different from those in the reference *w*Pip_Pel sequence. The pattern above the dotted lines represents the number of Illumina reads that have mapped for each position. The red box on the *w*Pip_Pel reference genome allows the IGV user to locate the zone that is visualized with more details on the bottom panels. **a** Mapping of the *cidA*/*cidB* operon reads onto the *w*Pip_Pel reference genome. Polymorphism between *w*Pip groups was detected (differences between Harash and Istanbul infected with *w*PipIV, Tunis, and *w*Pip_Pel infected with *w*PipI). Polymorphism was also detected within *Wolbachia* groups (variations observed between Tunis and *w*Pip_Pel or between Harash and Istanbul). The operon was also found to be polymorphic within the same isofemale line of *C. pipiens*. **b** Mapping of the *cinA*/*cinB* operon reads onto the *w*Pip_Pel reference genome. No polymorphism was detected between reads from the three isofemale lines and the reference line

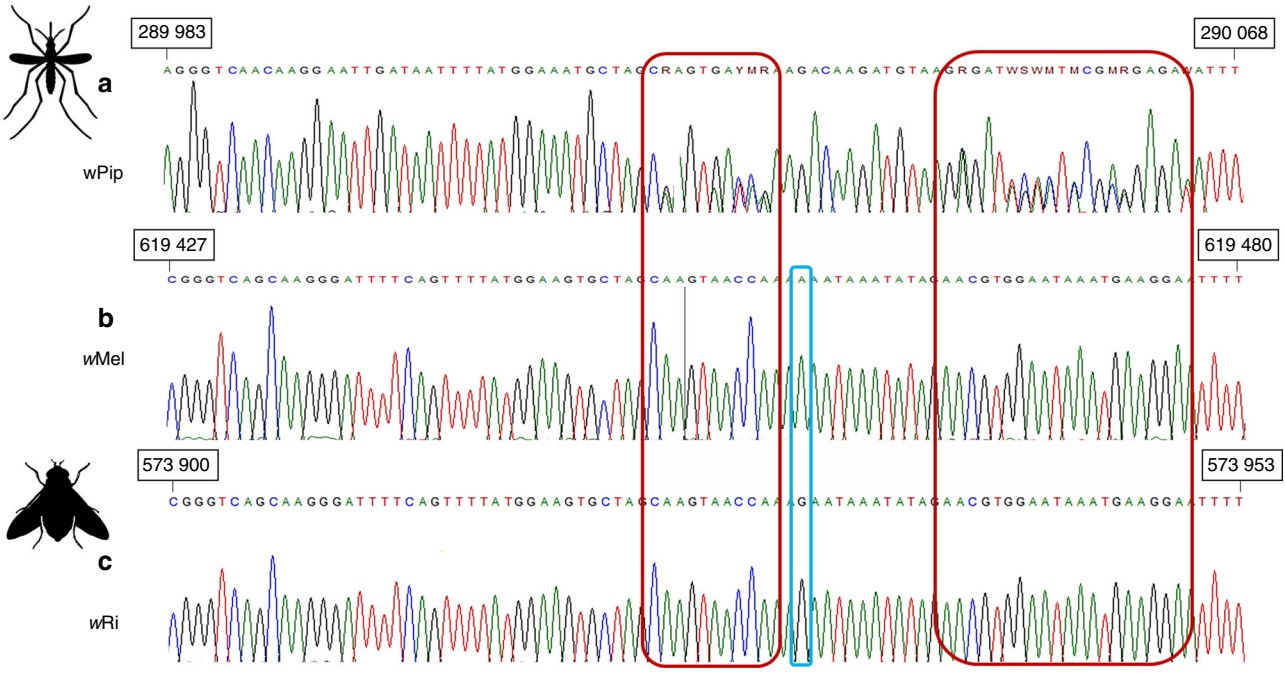

**Fig. 2** Sanger sequencing electrophoregrams of homologous regions of **a** *cidB* in *w*Pip, **b** *cifB* in *w*Mel, **c** and *cifB* in *w*Ri. The numbers in the black boxes indicate the positions of these regions in the *w*Mel, *w*Ri, and *w*Pip_Pel genomes. The blue rectangle highlights the only polymorphism between *w*Ri and *w*Mel. Unlike the *Wolbachia* strains of *Drosophila*, *w*Pip strains consistently give mixed signals (showed in red rectangles), suggesting the presence of at least two different sequences of this gene in DNA samples

**Table 1 Crossing types of the four mosquito lines each infected with *Wolbachia* *w*Pip from different groups (I, II, III, and IV)**

| ♂ \ ♀ | Tunis | Lavar | Maclo | Istanbul | Crossing type |
|---|---|---|---|---|---|
| Tunis (*w*PipI) | + | − | + | − | A |
| Lavar (*w*PipII) | + | + | + | − | B |
| Maclo (*w*PipIII) | + | + | + | + | C |
| Istanbul (*w*PipIV) | − | − | − | + | D |

+ means that the cross is compatible (many larvae hatch from the egg rafts) and − indicates an incompatible cross (no larvae hatch from the egg rafts). Each isofemale line has a different crossing type, as deduced from reciprocal crosses between the four lines.

variations. These variations occurred only in specific domains for both proteins, which consist exclusively of protein–protein interaction motifs and thus might be involved in the physical interaction between CidA and CidB.

## Results

**Illumina sequencing of several *w*Pip strains**. We first used Illumina technology to sequence three *w*Pip genomes from strains inducing different crossing types[13]: two strains from group IV,

originating from Algeria and Turkey; and one strain from group I, originating from Tunisia. The mapping of the reads onto the reference genome *w*Pip_Pel[18] revealed an abnormally high coverage of the WOPip1 region, which includes the *cidA* and *cidB* genes. The coverage for this region was, on average, three times greater than that for 14 single-copy housekeeping genes[19, 20] (Supplementary Fig. 1). Moreover, the *cidA-cidB* operon was highly variable between and within the three isofemale lines sequenced (Fig. 1a). Contrary to the assembly of *w*Pip_Pel

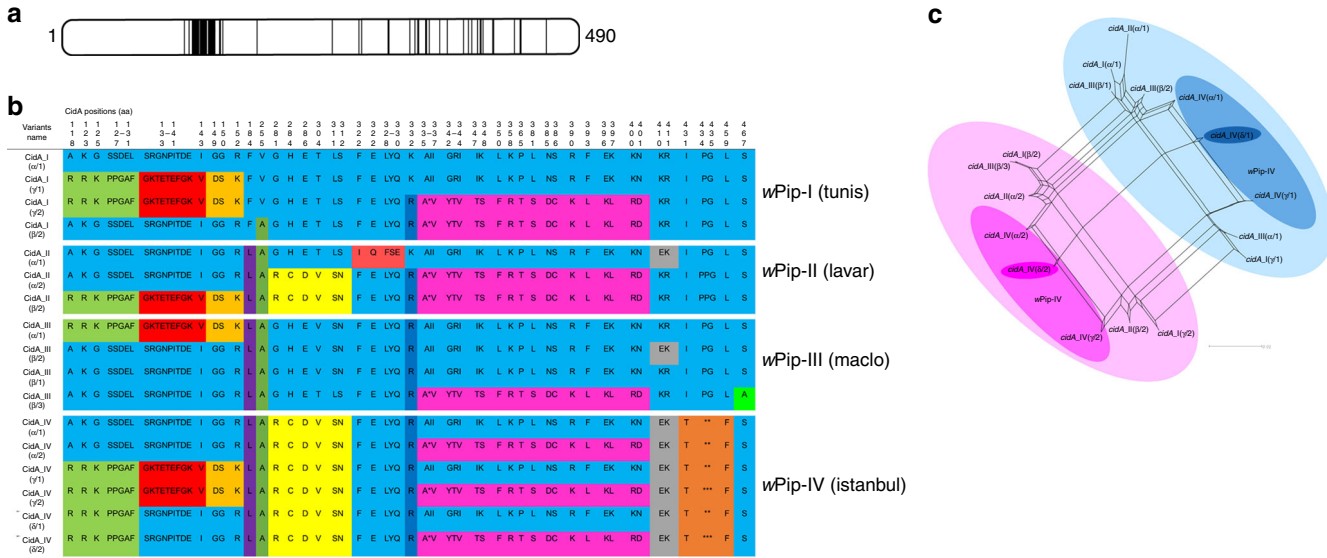

**Fig. 3** The repertoire of CidA variants in the different *w*Pip groups. **a** Localization of polymorphic zones within CidA proteins. Schematic representation of the CidA protein with polymorphic zones between CidA variants of the four *w*Pip groups highlighted in black. **b** The polymorphism between CidA variants is distributed as blocks of variable amino acids. Protein sequences alignment of the CidA variants found in the four *Wolbachia* strains Tunis, Lavar, Maclo, and Istanbul. The first sequence is used as a reference to determine the polymorphic region. For more clarity, only polymorphic positions of the alignment are represented and amino-acid positions are not contiguous. When more than two contiguous amino acids were variable the "-" symbol was used between the first and the last variable position of the zone. Colors show polymorphic blocks of residues present in variants regardless of their phylogenetic *w*Pip group (I–IV). However, no variant (i.e., complete CidA sequence) is common to *w*Pip strains from different groups. **c** *cidA* variants result from block recombination. Each edge (or set of parallel edges) corresponds to a split in the data set and has a length equal to the weight of the split. Incompatible splits produced by recombination are represented by boxes in the network. Most of the *cidA* variants are connected by multiple pathways resulting from block recombination between them. The largest circles highlight the two groups of *cidA* variants sharing the same amino-acid sequence from position 336 to 401 (**b**). Intermediate darker circles highlight *cidA* variants observed in the *w*PipIV group. The smallest bold circles highlight the two versions (1 and 2) of the *cidA*_IV(δ) variant matching the "incompatible" crossing type of mosquito lines infected with *w*PipIV (see the text)

genome, which exhibits only one copy of *cidA-cidB* operon[18], our findings suggest the occurrence of several copies of the *cidA-cidB* operon within each *w*Pip genome from Algeria, Tunisia, and Turkey. Consistent with this observation, the Sanger sequences of *cidA* and *cidB* obtained from single individuals displayed multiple overlaps (Fig. 2a). No such diversity was found in *w*Mel and *w*Ri from *D. melanogaster* and *D. simulans* (Figs. 2b, c). The *cinA* and *cinB* paralogs, which are also present in all sequenced *w*Pip genomes and have been identified as possibly involved in incompatibility between *C. pipiens* lines[10], displayed no polymorphism within and between *w*Pip strains (Fig. 1b), ruling out their role in CI diversity[10]. These findings suggest a previously unsuspected diversity of *cidA-cidB* copies within *w*Pip genomes, resulting from gene amplifications and divergence that may account for CI diversification in *C. pipiens*.

**High *cidA-cidB* diversity between *w*Pip groups**. We explored the association between *cidA-cidB* genetic variants and CI diversity in *C. pipiens* further, by investigating the variability of these genes in four isofemale lines differing by their geographical origin, each infected with a *w*Pip strain from a different group (I–IV) and exhibiting a different crossing type (Table 1). Each of these isofemale lines was founded with one initial egg-raft from a single female. Cloning and sequencing revealed a large number of *cidA-cidB* variants in each of these *w*Pip strains. The variable domains that differ between variants were not contiguous but restricted to some specific zones of the proteins (Figs. 3a and 4a). Within a single line there were up to six different variants of *cidA* (Fig. 3b), and up to four different variants of *cidB* (Fig. 4b). Some specificity in the *cidA-cidB* repertoire was also detected, with none of the variants shared between strains from different *w*Pip

groups (Figs. 3b and 4b). Network analyses of *cidA* and *cidB* suggested that the differences between copies stemmed mostly from block rearrangements within and/or between *w*Pip genomes (Figs. 3c and 4c). These rearrangements were never observed in the region corresponding to the CidB catalytic domain, but in other regions of the CidA and CidB proteins (Figs. 3 and 4; Supplementary Fig. 2 and 3). The stability of the *cidA-cidB* repertoire over a decade (between 2006 and 2017) in addition with the stability in the *cidA* and *cidB* copy numbers relative to *wsp* gene (known to be present as a single copy in all *Wolbachia* genomes) inferred by quantitative PCR (q-PCR) in the Istanbul *C. pipiens* line are convincing elements that the *cidA-cidB* genes are present in multiple copies in *w*Pip genome. The copy number ratio between *cidA-cidB* genes and *wsp* was always between 4 and 6 (Table 2). We also found that all copies of these genes were transcribed. Indeed, in the *w*Pip strain from Istanbul the transcripts of the six polymorphic copies (i.e., variants) of *cidA* and the four polymorphic copies of *cidB* were detected in RNA extract after reverse transcription-PCR, cloning, and sequencing.

**cidA-cidB repertoire variations are linked to crossing type**. We then focused on a simpler situation, by analyzing CI diversity within *w*Pip group IV. We used six isofemale lines with two different crossing types. These lines had the same rescue properties but two different mods[13]. Males from "incompatible" lines sterilize females whenever they host a *w*Pip group I, II, or III strain, whereas males from "compatible" lines always produce progenies with these females (Fig. 5). We compared the *cidA-cidB* repertoires of "compatible" and "incompatible" lines. We found that one CidA variant [*cidA*_IV(α)] was present in all *w*PipIV lines, regardless of their crossing type or geographic origin

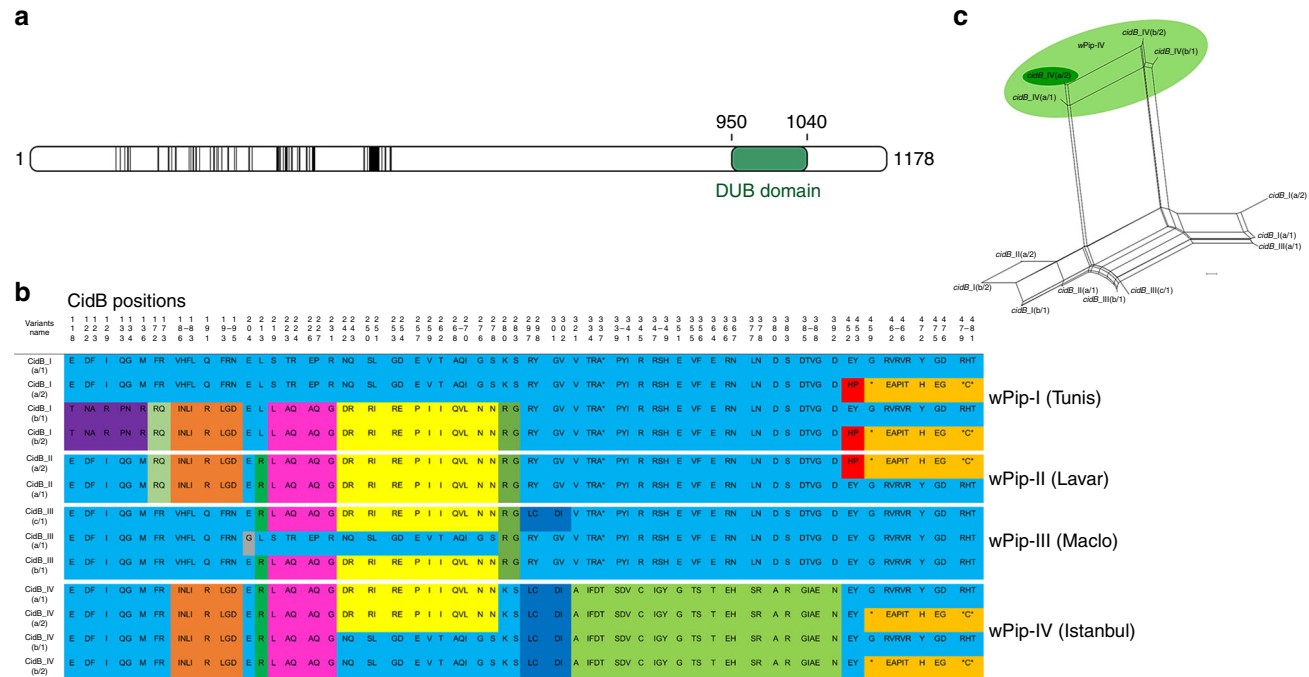

**Fig. 4** The repertoire of CidB variants in the different *w*Pip groups. **a** Localization of polymorphic zones within CidB proteins. Schematic representation of the CidB protein with polymorphic zones between CidB variants of the four *w*Pip groups highlighted in black. The deubiquitylating protease domain (DUB) present in CidB, represented in green, displayed no polymorphism between *w*Pip strains. **b** The polymorphism between CidB variants is distributed as blocks of variable amino acids. Protein sequences alignment of the CidB variants found in the four *Wolbachia* strains Tunis, Lavar, Maclo, and Istanbul. The first sequence is used as reference to determine the polymorphic region. For more clarity, only polymorphic positions of the alignment are represented and amino-acid positions are not contiguous. When more than two contiguous amino acids were variable the "-" symbol was used between the first and the last variable position of the zone. Colors show polymorphic blocks of residues present in variants regardless of their phylogenetic *w*Pip group (I–IV). However, no full variant (i.e., complete CidB sequence) is common to *w*Pip strains from different groups. **c** *cidB* variants result from block recombination. Each edge (or set of parallel edges) corresponds to a split in the data set and has length equal to the weight of the split. Incompatible splits produced by recombination are represented by boxes in the network. Most of the *cidB* variants are connected by multiple pathways resulting from block recombination between them. The light green circle highlights *cidB*_IV variants, which cluster together due to their similar amino-acid sequences from position 321 to position 392 (**b**). The *cidB*_IV(a/2) variant matching the "incompatible" crossing type of mosquito lines infected with *w*PipIV (see the text) is highlighted by the darker green circle

**Table 2 Quantification of *cidA-cidB* operon in different *w*Pip genomes**

| Line / Ratio | Tunis / *w*Pip-I | Lavar / *w*Pip-II | Maclo / *w*Pip-III | Istanbul / *w*Pip-IV |
|---|---|---|---|---|
| *cidA/wsp* | 5,99 ± 0,77 | 5,40 ± 0,84 | 5,1 ± 0,41 | 5,62 ± 0,72 |
| *cidB/wsp* | 6,21 ± 1,40 | 5,06 ± 1,46 | 4,17 ± 0,64 | 6,13 ± 1,26 |
| *cidA/cidB* | 0,99 ± 0,21 | 1,10 ± 0,19 | 1,24 ± 0,21 | 0,95 ± 0,19 |

The number of copies of the *cidA* and *cidB* genes was estimated by real-time q-PCR, relative to the reference gene *wsp* known to be present as a single copy per genome. The number of copies was determined by calculating the ratio between *cidA* (or *cidB*) signals and *wsp* signals on four individuals per line. ± means s.d.The *cidA* to *cidB* ratio was always close to one. The numbers of *cidA* and *cidB* copies present per *Wolbachia* genome were therefore similar, as expected for an operon.

(Fig. 5). Such a ubiquitous distribution is expected for the resc factor mediating compatibility between all isofemale lines hosting *w*Pip strains from the same group. By contrast, no CidB variant was shared between compatible and incompatible lines (Fig. 5). Strikingly, *cidA* and *cidB* repertoire variations were clearly associated with crossing type variations, as *cidA*_IV(δ) and *cidB*_IV

(a/2) variants were found uniquely in "incompatible" lines (Fig. 5). This may suggest that both *cidA*_IV(δ) and *cidB*_IV(a/2) are required for the incompatibility phenotype (Fig. 5).

We further investigated the distribution of these two specific *cidA/cidB* variants [*cidA*_IV(δ) and *cidB*_IV(a/2)] at a larger scale, by performing diagnostic restricted fragment length

**a**

| Line name | Crossing type | Polymorphism location | | *cidA_IV* | | | |
|---|---|---|---|---|---|---|---|
| | | Upstream | | α | β | γ | δ |
| | | Downstream | | 1/2 | 1/2 | 1/2 | 1/2 |
| Istanbul | Incompatible | | | P | A | P | P |
| Hang Zhou | Incompatible | | | P | A | A/P | P |
| Ichkeul 09 | Incompatible | | | P | A | A | P |
| Ichkeul 21 | Incompatible | | | P | A | A | P |
| Harash | Compatible | | | P | P | A | A |
| Ichkeul 13 | Compatible | | | P | P/A | A | A |

**b**

| Line name | Crossing type | *cidB_IV* | | | | | |
|---|---|---|---|---|---|---|---|
| | | a1 | a2 | a3 | b1 | b2 | b3 |
| Istanbul | Incompatible | P | P | A | P | P | A |
| Hang Zhou | Incompatible | P | P | P | P | A | P |
| Ichkeul 09 | Incompatible | P | P | A | A | A | A |
| Ichkeul 21 | Incompatible | P | P | P | P | A | P |
| Harash | Compatible | P | A | P | P | A | P |
| Ichkeul 13 | Compatible | A | A | A | P | A | P |

**Fig. 5** *cidA*_IV(δ) and *cidB*_IV(a/2) are present specifically in lines infected with *w*PipIV strains displaying the "incompatible" crossing type. The "incompatible" or "compatible" crossing type of the *C. pipiens* lines infected with *w*PipIV strains were determined by crossing males with females infected with *w*PipI, *w*PipII, or *w*PipIII. A "compatible" type corresponds to a line in which all males are compatible when crossed with all tested females, and an "incompatible" type corresponds to a line in which males are incompatible when crossed with all tested females. The column of the two variants *cidA* IV(δ) and *cidB*_IV(a/2) associated with the "incompatible" crossing type were shaded in green. **a** Distribution of *cidA* variants (P for present, A for absent) in six *w*PipIV strains. Each column corresponds to the specific upstream polymorphic region named α, β, γ, or δ. The downstream polymorphism is represented by either the sequence 1 or the sequence 2 (1/2). For a given upstream sequence P/A means that the strain exhibits the sequence 1 but not the sequence 2. On the opposite A/P means that the strain exhibits the sequence 2 but not the sequence 1. **b** Distribution of *cidB* variants (P for present, A for absent) in six *w*PipIV strains

polymorphism (RFLP) tests on 180 isofemale lines from 15 populations [13 in Algeria and Tunisia; 1 each in China and Turkey], for which we determined crossing types. A very strong association was established between the presence of both *cidA*_IV (δ) and *cidB*_IV(a/2) in "incompatible" isofemale lines and their absence from "compatible" isofemale lines: all 17 "incompatible" lines had both these variants, whereas 147 of the 163 "compatible" lines had neither of them ($\chi^2 = 78$, df = 1, $p < 2.2e\text{-}16$). Among the 16 "compatible" isofemale that appeared discordant: (i) 8 were truly discordant as they display both *cidA*_IV(δ) and *cidB*_IV(a/2) while being compatible but (ii) 8 were only discordant for *cidA* as they displayed *cidA*_IV(δ) but lacked *cidB*_IV(a/2) (as confirmed by cloning and sequencing; Supplementary Data 1). To summarize the data: the absence of these variants is always associated with a "compatible" crossing type while its presence is mostly associated with "incompatible" crossing type with few discrepancies.

**Putative ankyrin interaction domains between CidA and CidB.** The analysis of CidA and CidB polymorphic regions revealed that the sequence variations associated with crossing type variations are restricted to about 30 amino-acid positions, for both proteins (Fig. 6a). Extensive fold-recognition analysis of the CidA and CidB proteins were performed using the @tome2 in-house server[21]. Except for the DUB protease domain, no tridimensional model could be firmly obtained. Nevertheless, both CidA and CidB exhibit sequence identity homology (under 20%) with protein–protein interaction domains with repeated structure. Careful examination of hydrophobic patterns suggests internal helical structure repetitions as found in ankyrin or HEAT repeats[22]. The size of ~33 amino acids is the universal expected

size for tandem ankyrin repeat domains[6]. However, due to the lack of available homolog sequences in databases, it is difficult to determine the helical structure repetition boundaries in both N- and C-terminal part of the protein domains. As the protein variations matching with crossing type variations occur in putative ankyrin domains, we hypothesize that CidA and CidB co-diversify in these domains because they are involved in their reciprocal interactions (Fig. 6).

**Discussion**

In the *C. pipiens* mosquitoes, the genetic architecture of the *Wolbachia cidA-cidB* operon is characterized by variable multi-copies associated to the unique CI diversification observed in this species. These different copies of *cidA* and *cidB* likely result from genic amplification followed by diversification since we revealed a high diversity within each *w*Pip strain. As these genes are associated with the prophage WO modules[11], the number of *cidA-cidB* copies assessed by q-PCR may not reflect the exact number of *cidA-cidB* copies in a single *w*Pip genome since viral particle multiplication resulting from lytic phage activity can influence the accuracy of this quantification[23]. However, the diversity of *cidA-cidB* sequences in a single mosquito, the geographical and temporal stability of sequences found in *w*PipIV group clearly point out that several copies of the operon do co-exist within the same *w*Pip genome. As *cidA* and *cidB* are amplified in all *w*Pip genomes from each *C. pipiens* individual we sequenced so far, the presence of only one copy of the *cidA/cidB* operon in *w*Pip_Pel seems to result from an error during assembly likely caused by too many repeats. The reason(s) why such amplification and diversification of *cidA/cidB* has evolved in *w*Pip is still an open

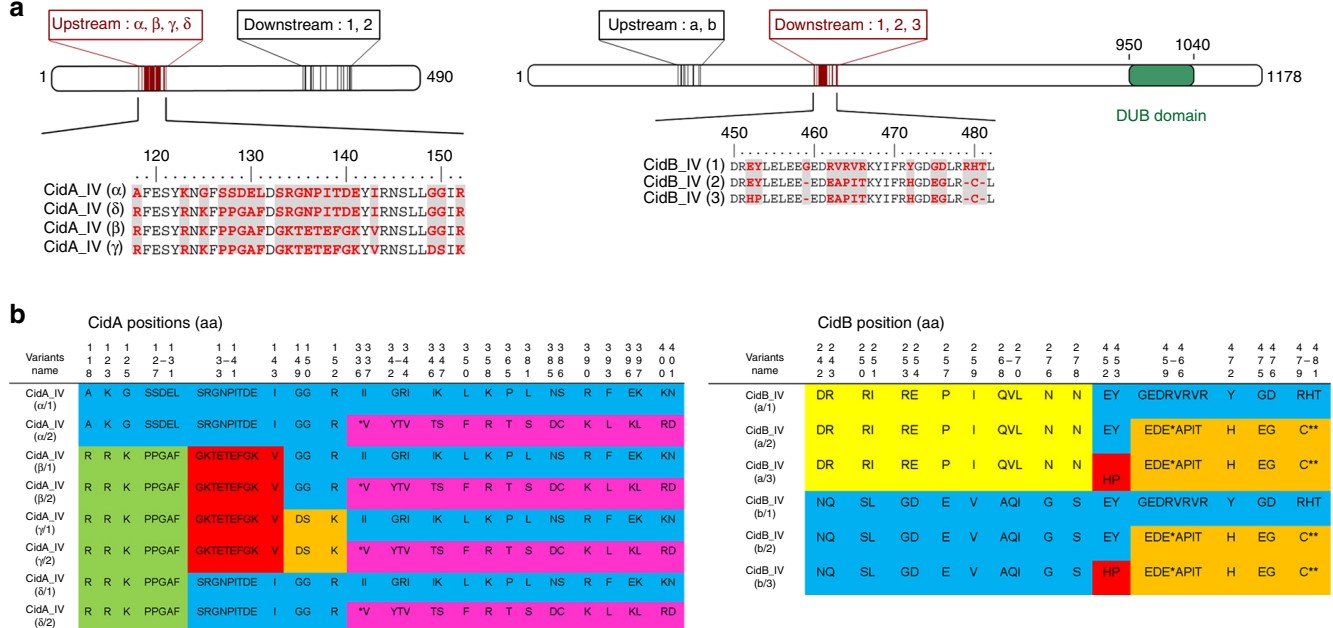

**Fig. 6** The putative domains of interaction between the CidA_IV and CidB_IV proteins. **a** Localization of polymorphic zones within CidA_IV and CidB_IV proteins. Secondary structure prediction and fold-recognition analysis predicted that the CidA protein could almost exclusively display protein/protein interaction repeated motifs as ankyrin- (or HEAT-like repeats). Similar domains of ~30 amino acids (expected size for ankyrin domains) are also present in CidB. The deubiquitylating protease domain (DUB) present in CidB, represented in green, displayed no polymorphism between wPipIV strains as already observed between wPip strains from different groups. The polymorphism of the wPipIV CidA and CidB variants associated with crossing type ("compatible" vs. "incompatible", see Fig. 5 for definition) is highlighted in red. The corresponding variations in protein sequences are reported below the schematic representation of the full-length proteins, with the variable residues highlighted in red. Other sequence variations not predicted to be involved in CI variations are shown in black. **b** CidA_IV and CidB_IV variants sequences detected in all sequenced wPip strains from group IV. CidA_IV variants display two regions of polymorphism resulting from recombination revealed by block colors: the upstream one from 118 to 152 aa and the downstream one from 336 to 401 aa. Four possible sequences (noted α, β, γ, or δ) were found in the upstream polymorphic region followed by one of the two sequences in the downstream region (noted 1 or 2). Only the upstream polymorphic region of CidA is associated with crossing type variations. CidB_IV variants display two regions of polymorphism resulting from recombination: the upstream one from 242 to 278 aa and the second one from 450 to 481 aa. Two possible sequences (noted a and b) were found in the upstream polymorphic region followed by one of the three possible sequences in the downstream polymorphic region (noted 1, 2, and 3). Only the downstream region is associated with crossing type variations

question. However, this diversity likely reflects high levels of intra- or intergenome recombination, which may be favored by multiple infections with different wPip and/or promoted by lytic phage activity in *C. pipiens*[24].

In a more global context, our results provide insights into the respective roles of CidA and CidB in the mod/resc model. Indeed, the association between *cidA*_IV(δ)-*cidB*_IV(a/2) presence/absence and incompatible/compatible crossing types, together with the results of functional studies[10, 11], demonstrate that CidB is involved in the mod function responsible for the variation in crossing types. However, some discrepancies were found in few isofemale lines, which harbor *cidA*_IV(δ) and/or *cidB*_IV(a/2) but are "compatible". These discrepancies suggest that either (i) our strategy, involving larval field sampling, the establishment of isofemale lines and crossing experiments, generated a few errors or (ii) other genetic or epigenetic mechanisms may disrupt the relationship between genomic potential and the resulting crossing type.

The role of CidA is still unclear. The association of *cidA*_IV(δ) with variation in crossing types does not necessarily implicate CidA in the mod function. Indeed, as CidA and CidB interact with each other[10], their co-diversification could result in specific CidA variations whenever involved in the mod (i.e., toxin) or in resc (i.e., antitoxin) function. Several lines of evidence suggest that CidA could be the antitoxin of CidB and that the *cidA-cidB* operon encodes a classic toxin–antitoxin system: (i) the simple two-gene structure of the operon, with the putative antitoxin gene located upstream from the putative toxin gene[9]; (ii) the over-expression of the antitoxin gene relative to the toxin gene[11]; (iii) the ability of the two proteins to form a complex[10]; and (iv) the required co-expression of *cidA* and *cidB* for the production of live transgenic *D. melanogaster* and *Saccharomyces cerevisiae*[10]. Additional argument of CidA being an antitoxin is provided by the presence of *cidA*_IV(α), a ubiquitous CidA variant in wPip strains within the same group wPipIV, potentially accounting for their hosts' compatibility. Moreover, if CidA is indeed the antitoxin, then a new toxin (i.e., CidB) produced by block recombination between variants would be positively selected only if a compatible CidA, capable of interaction with this new toxin, was already present in the same wPip genome to prevent its producer from being killed[2, 25]. This is coherent with our results since we found more *cidA* variants than *cidB* variants in the same isofemale line (except for Tunis, which has as many *cidA* as *cidB*).

Most toxin–antitoxin systems require an interaction between both proteins[25]. Pull-down experiments have demonstrated that CidA can interact with CidB[10]. Our analysis of CidA-CidB repertoires in wPipIV group showed that genetic variations linked to crossing type variations occurred only in specific domains of about ~30 amino acids in length, for both proteins (Fig. 6a). These domains consist exclusively of protein–protein interaction motifs (ankyrin or HEAT repeats) and thus might be involved in the reciprocal interaction between CidA and CidB. The variability of these domains that co-diversify may account for specific interaction between one variant of CidA and one variant of CidB.

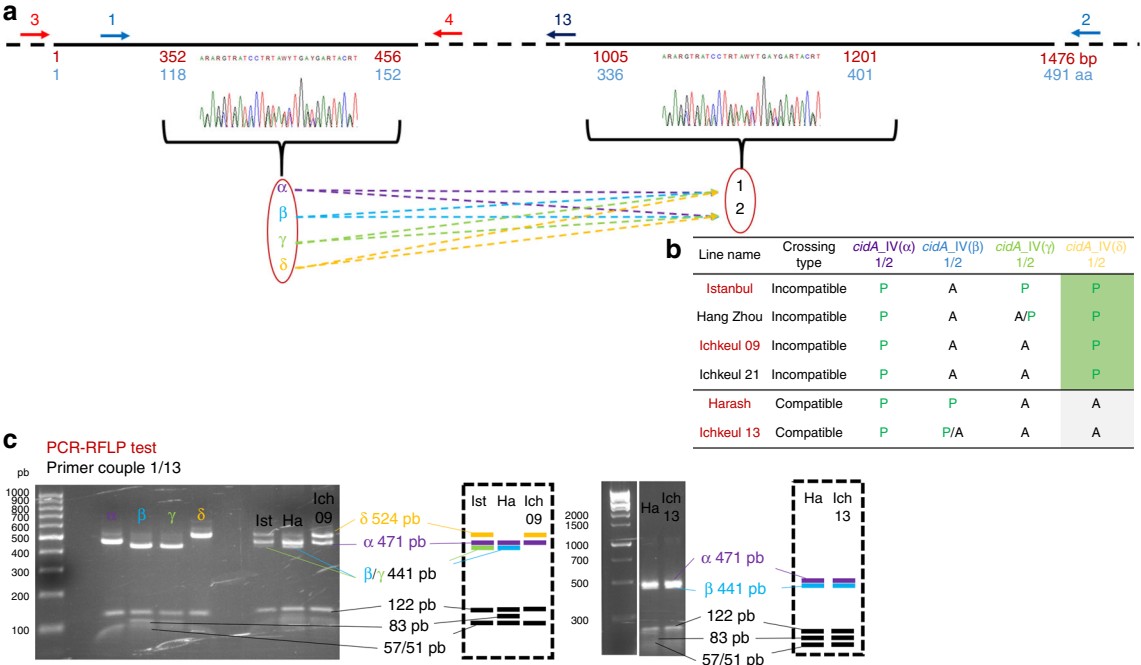

**Fig. 7** *cidA*_IV gene polymorphic regions and PCR-RFLP tests for specific variants. **a** Schematic representation of the architecture of *cidA*_IV polymorphism. The black line represents the *cidA*_IV sequence with a focus on the two polymorphic regions. Non-polymorphic regions were shortened and represented as a dashed line. Numbers in red under the line represent nucleotide positions, and the numbers in blue indicate amino-acid positions. The overlapping oligonucleotides used for PCR amplification are represented by arrows numbered as in Supplementary Table 2. Both polymorphic regions were studied, but only the upstream matched the compatibility profile; its four different sequences (α, β γ, and δ) were followed by one of the two possible sequences (1 or 2) in the downstream polymorphic region. A different color code was used for α (purple), β (light blue), γ (light green), and δ (yellow) sequences in the upstream part of the *cidA* gene. **b** The repertoire of *cidA*_IV variants is different in "compatible" and "incompatible" lines. *cidA*_IV(δ) is present only in lines with "incompatible" crossing type. The names of the *C. pipiens* lines used to set-up the PCR-RFLP (**c**) test are highlighted in red. **c** PCR-RFLP tests for distinguishing between *cidA*_IV variants on the basis of the upstream polymorphic region. A 778 bp fragment was amplified with primers 1/13. Double digestion with *Apo*I and *Hpy*188I distinguished between *cidA*_IV(α) (six fragments: 471; 122; 57; 53; 51; and 24 bp), *cidA*_IV(β) and *cidA*_IV(γ) (six fragments: 441; 122; 83; 57; 51; and 24 bp), and *cidA*_IV(δ) (five fragments: 524; 122; 57; 51; and 24 bp). Panel to the left of the electrophoresis gel: PCR-RFLP on DNA from clones; right panel: PCR-RFLP on DNA from the Istanbul, Harash, Ichkeul 13, and Ichkeul 09 lines. On the right of the gel, a schematic representation of the PCR-RFLP profiles of these lines. Digestion bands that are specific of the variants are represented with the color code established in **a**. Bands that are not used to discriminate between the variants are represented in black on the bottom of the schematic gel

Such variability in interactions between CidA and CidB could drive the unrivaled diversity of crossing types in *C. pipiens*.

## Methods

**Isofemale line construction.** *C. pipiens* larvae and pupae were collected in the field and reared to adulthood in the laboratory. Females were then fed on blood to lay eggs that served to establish isofemale lines. Each egg-raft (containing 100–300 eggs) was individually isolated for hatching, and the *Wolbachia* group present was determined by performing *pk*1 PCR-RFLP tests[26] on two first-instar larvae (L1). Isofemale lines were created by rearing the offspring resulting from a single egg-raft (thus from a single female). We established 162 isofemale lines for this study. We also used 21 isofemale lines from laboratory stocks of various geographic origins (Supplementary Data 1 and Supplementary Table 1). Isofemale lines were reared in 65 dm³ screened cages kept in a single room at 22–25 °C, under a 12 h light/12 h dark cycle. Larvae were fed with a mixture of shrimp powder and rabbit pellets, and adults were fed on honey solution.

**Crossing type determination.** Crossing types were characterized by crossing males (25–50 virgin males) from each of the studied isofemale lines with females (25–50 virgin females) from four reference laboratory isofemale lines hosting different *Wolbachia* strains [Tunis (*w*PipI), Lavar (*w*PipII), Maclo (*w*PipIII), and Istanbul (*w*PipIV)] or with females from Tunis only (Supplementary Data 1 and Supplementary Table 1).

**Illumina sequencing.** The genomes of three *Wolbachia* strains from three *C. pipiens* isofemale lines were fully sequenced: Tunis from group *w*PipI, and Harash and Istanbul from group *w*PipIV. *Wolbachia* is an intracellular uncultivable bacterium. To obtain satisfactory amounts of *Wolbachia* DNA, we therefore developed a protocol based on that described by Ellegaard et al.[27]. This protocol involved *Wolbachia* enrichment on egg rafts immediately after oviposition, to maximize the

ratio of *Wolbachia* to *Culex* genomes. The freshly laid eggs-rafts were washed in bleach to crack their chorion. They were then rinsed in water and homogenized in phosphate-buffered saline (1× PBS). *Wolbachia* cells were isolated from each egg-raft separately and concentrated. Each egg-raft was crushed with a sterile pestle and the resulting suspension was centrifuged at 400 × *g* for 5 min at 4 °C to remove the cell debris. The supernatant was then centrifuged again at 6000 × *g* for 5 min at 4 °C, to obtain a pellet of *Wolbachia* cells. These cells were resuspended in 1× PBS and passed through a filter with 5 μm pores (GVS Filter Technology), and a filter with 2.7 μm pores (Whatman), to remove any remaining particles bigger than *Wolbachia*. The resulting filtrate was centrifuged at 6900 × *g* for 15 min at 4 °C to obtain a pellet of *Wolbachia* cells. Multiple-displacement amplification was carried out directly on the *Wolbachia* pellet, with the Repli-g- mini kit (Qiagen). For each strain, five amplifications were performed independently, to randomize potential amplification errors, and the amplicons were then pooled for sequencing. Illumina sequencing was performed by the 3 kb Long Jumping Distance Mate-Pair method, on a HiSeq2000 instrument (Eurofins MWG Operon platform in Ebersberg, Germany), generating 2 × 100 bp sequences for each fragment.

**Cloning and Sanger sequencing of the *cidA* and *cidB* genes.** For all experiments, total DNA was extracted with the Dneasy Blood & Tissue Spin-Column kit (Qiagen; bench protocol: animal tissues). For amplification of the full-length *cidA* gene, we used two pairs of overlapping primers, 1/2 and 3/4 (Fig. 7, Supplementary Table 2). For amplification of the full-length *cidB* gene, we used four pairs of overlapping primers: 5/6; 7/8; 9/10; and 11/12 (Fig. 8, Supplementary Table 2). We used *cidA* fragments amplified with primers 1/2 and *cidB* fragments amplified with primers 7/8 for cloning experiments, because these two regions contain all the polymorphism observed in *cidA* and *cidB* variants (Figs. 3 and 4). PCR products were extracted from the electrophoresis gel with the W-tracta disposable gel extraction tool (Star Lab) and purified with the QIAquick Gel Extraction Kit (Qiagen). The TOPO TA cloning Kit pCR 2.1-TOPO Vector (Invitrogen) was used for ligation. Before electroporation, the DNA was dialyzed with VSWP membrane

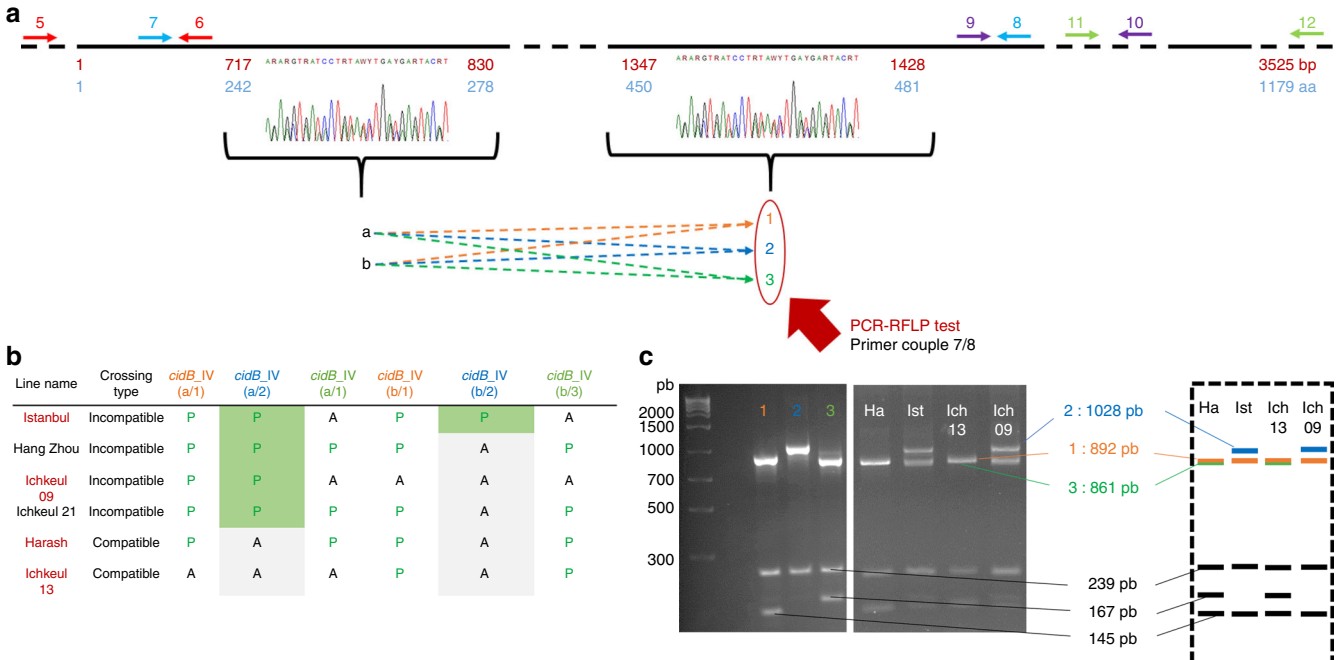

**Fig. 8** *cidB*_IV gene polymorphic regions and RFLP tests for specific variants. **a** Schematic representation of the polymorphism architecture of *cidB*_IV. The black line represents the *cidB*_IV sequence with a focus on the polymorphic region. Non-polymorphic regions were shortened and represented as a dashed line. Numbers in red represent nucleotide positions and those in blue indicate amino-acid positions. The overlapping oligonucleotides used for amplification are represented by arrows numbered as in Supplementary Table 2. Both polymorphic regions are indicated but only the downstream (on the left, base pairs 1347–1428, amino acids 450–481) matches the compatibility profiles. The two sequences of the upstream polymorphic zone (a and b) are followed by one of the three possible sequences (1, 2, and 3) in the downstream polymorphic region. A different color code was used for 1 (orange), 2 (blue), and 3 (green) different sequences in the downstream part of the *cidB* gene. **b** The *cidB*_IV variant repertoire differs between "compatible" and "incompatible" lines. *cidB*_IV(a/2) is present only in lines with "incompatible" crossing type. The names of the *C. pipiens* lines used to set up the PCR-RFLP (**c**) test are highlighted in red. **c** PCR-RFLP tests for distinguishing between *cidB*_IV variants on the basis of the second polymorphic region. A 1267–1276 bp fragment was amplified with primers 7/8. Double digestion with *Ban*I and *Taqa*I distinguished between *cidB*_IV(1) (three fragments: 892; 239; and 145 bp), *cidB*_IV (2) (two fragments: 1028 and 239 bp), and *cidB*(3) (three fragments: 861; 239; and 167 bp). Panel to the left of the electrophoresis gel: PCR-RFLP on DNA from clones; right panel: PCR-RFLP on DNA from lines. The panel on the right shows schematic representations of the PCR-RFLP profiles for these lines. Digestion bands that are specific of the variants are represented with the color code established in the **a**. Bands that are not used in to discriminate between the variants are represented in black on the bottom of the schematic gel

filters with 0.025 μm pores (Merck Millipore) to remove excess salt from the ligation. Electroporation was performed with One Shot TOP10 Electrocomp *Escherichia coli* (Invitrogen) and a Bio-Rad Micropulser with 2 mm-path electroporation cuvettes (Eurogentec) according to the manufacturer's instructions. For repertoire acquisition of *cidA* and *cidB*, at least 48 clones for each individual were amplified and sequenced with M13 primers. Once the PCR-RFLP tests (see below) had been developed, more sequences were obtained for the variants initially underrepresented. For all the *C. pipiens* isofemale lines analyzed by cloning-sequencing, direct Sanger sequencing results for *cidA* and *cidB* were compared with the repertoire obtained by cloning-sequencing, to ensure that no variants were missing. For Sanger sequencing, the PCR products were purified with the Agencourt Ampure PCR purification kit (Agencourt) and directly sequenced with an ABI Prism 3130 sequencer using the BigDye Terminator Kit (Applied Biosystems).

**Sanger sequencing data analyses.** Sequence variants of *cidA* and *cidB* were aligned, with Muscle implemented in Seaview 6.4.1 software[28]. Alignments were then analyzed within a phylogenetic network framework, to account for potentially conflicting signals due to recombination. A phylogenetic network was constructed from uncorrected *P* distances by the neighbor-net method[29] implemented in Splitstree4[30].

**PCR-RFLP tests.** We designed PCR-RFLP tests to detect the presence of the *cidA*_IV(δ) [*cidA*_IV(δ/1) and *cidA*_IV(δ/2)] and *cidB*_IV(2) [*cidB*_IV(a/2) and *cidB*_IV(b/2)] variants in isofemale lines. For *cidA*_IV variants, only the first polymorphic region was correlated with crossing types. We therefore designed the PCR-RFLP test to distinguish between the *cidA*_IV(α), *cidA*_IV(β), *cidA*_IV(γ), and *cidA*_IV(δ) sequences (Fig. 7). A 778 bp fragment was amplified with the 1/13 primer pair (Supplementary Table 2). Double digestion of the PCR products with *Apo*I and *Hpy*188I (New England Biolabs) identified three groups of sequencing: *cidA*_IV(α) (six fragments: 471; 122; 57; 53; 51; and 24 bp); *cidA*_IV(β) and

*cidA*_IV(γ) (six fragments: 441; 123; 83; 57; 51; and 24 bp); and *cidA*_IV(δ) (five fragments: 524; 122; 57; 51; and 24 bp).

For *cidB*_IV variants, only the second polymorphic region was found to be correlated with crossing types, so the PCR-RFLP test was designed to discriminate between the *cidB*_IV(1), *cidB*_IV(2), and *cidB*_IV(3) sequences (Fig. 8). A 1267–1276 bp fragment was amplified with the 7/8 primer pair (Supplementary Table 2). Double digestion of the PCR products with *Ban*I and *Taqa*I (New England Biolabs) distinguished three different groups of sequences: *cidB*_IV(1) (three fragments: 892, 239, 145 bp); *cidB*_IV(2) (two fragments: 1028 and 239 bp); and *cidB*_IV(3) (three fragments: 861; 239; and 167 bp).

**Gene copy number quantification.** The copy numbers of the *cidA* and *cidB* genes were estimated per *Wolbachia* genome, for four laboratory lines, on male and female adults and a pool of larvae at the earliest stage, by real-time q-PCR with a Roche Light Cycler. Three PCRs were performed, in triplicate, on each sample, one for the *Wolbachia wsp* locus present as single copy per genome (primer pair 18/19), one for *cidA* (primer pair 14/15), and one for *cidB* (primer pair 16/17). These primer pairs specifically bind to a region conserved in all *w*Pip groups for each gene (Supplementary Table 2). The ratio between *cidA* or *cidB* and *wsp* signals was used to estimate the relative number of copies of *cidA* or *cidB* per *Wolbachia* genome.

**cidA-cidB expressed repertoire.** Fresh mosquito samples from adult females and males and first (L1) and last (L4) instar larvae were used for RNA extraction with Trizol (Life Technologies). The RNA obtained was treated with DNase with the TURBO DNA-free Kit (Life Technologies), in accordance with the manufacturer's instructions. We reverse-transcribed 2–5 μg of each total RNA sample with the SuperScript III Reverse Transcriptase Kit and 30 ng of random oligomer primers ((RP)10; Invitrogen, Life Technologies).

The polymorphic zones of the *cidA* and *cidB* cDNAs were amplified, cloned and 48 clones were sequenced, as described above.

**Statistical analyses**. All statistical analyses were performed with R version 3.0.2 software. We compared the proportions of "compatible" and "incompatible" isofemales harboring $cidA$_IV($\delta$) and $cidB$_IV(a/2) in a two-sample test for equality of proportions with continuity correction based on $\chi^2$.

**Mapping of illumina reads on the reference genome**. Illumina Long Jumping Distance sequencing data were trimmed with Trimmomatic version 0.36.

Paired reads and single reads for each line were compiled into Fastq files and mapped onto the reference genome $w$Pip_Pel, with bwa mem and default parameters. Mappings were visualized with IGV version 2.3.68.

For each $Wolbachia$ strain, the coverages of WP0282 ($cidA$), WP0283 ($cidB$), WP0294 ($cinA$), WP0295 ($cinB$), and MLST genes[19, 31] were calculated with genomecov from bedtools, with the -d option, to calculate "per-base" genome coverage. Several adjustments were made: (i) coverage was normalized by dividing by the mean coverage on the whole reference genome of the strain concerned, to eliminate bias due to the higher coverage for strains for which there were larger numbers of reads in the Illumina data set; (ii) polymorphic regions of $cidA$ and $cidB$ were removed, due to the considerable variability of coverage in these regions; and (iii) the region of $cidB$ from position 2461 to the end of the gene was removed because this region is identically duplicated in the $w$Pip_Pel genome[9], artificially increasing its coverage.

**Data availability**. The Illumina raw reads data of the three $w$Pip genomes have been deposited in the Sequence Read Archive with Bioproject PRJNA393398 and BioSample SAMN07315580 ($w$PipI Tunis), SAMN07327402 ($w$PipIV Harash), and SAMN07327406 ($w$PipIV Istanbul). Nucleotidic and proteic sequences of $cidA$-$cidB$ variants have been deposited in GenBank and accession numbers are available in the Supplementary Table 3. The authors declare that all other data supporting the findings of this study are available within the article and its Supplementary Information files, or are available from the authors upon request.

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

## Acknowledgements

We would like to thank Sylvain Charlat, Pierrick Labbé, Benjamin Lopin, Nicole Pasteur, and François Rousset for helpful comments on the manuscript. We also thank Khalid Belkhir for bioinformatic analyses performed on the Montpellier Bioinformatics Biodiversity cluster computing platform. Frédéric Delsuc for phylogenic analyses, and Arnaud Berthomieu, Ali Bouattour, Emilie Dumas, Patrick Makoundou, and Sandra Unal for technical support. Sequencing data were produced through the technical facilities of the LabEX "Centre Méditerranéen de l'Environnement et de la Biodiversité" in the GENSEQ plateform. This work was funded by the French ANR (project "CIAWOL" ANR-16-CE02-0006-01 and ANR-10-BINF-0003).

## Author contributions

M.B., M.S., and M.W. conceptualized and designed the study; M.B., C.A., MaBe, and F.J. performed the experiments; M.B., M.C.-G., M.S., and M.W. analyzed and interpreted the data; M.B., M.S., and M.W. wrote the manuscript.

## Additional information

**Competing interests:** The authors declare no competing financial interests.

