## [Peer Review File · Nature Communications]

Reviewers' comments:

Reviewer #1 (Remarks to the Author):

Summary: This is an excellent study exploring the diversity of the *cidA-cidB* genes in both copy number and sequence, using the model CI system *Culex pipiens*. The research is very well done, from documenting *Wolbachia* strain variation within the species, to investigating CI differences in controlled genetic backgrounds. The logic flow of some of the statements in the abstract and introduction needs improvement, apparently because the authors are anticipating their own results (e.g. copy number differences and patterns of variation in *cidA* and *cidB*). Also, there are some missing references that should be added. I anticipate that this paper will be a significant contribution to our understanding of the genetic basis of CI variation in nature. Statements about "domains of interaction" between *cidA* and *cidB* are hypotheses. But it is described in the abstract as if it is already known that these proteins interact directly with each other, although it is not known. More caution in interpreting these results, and consideration of alternate hypotheses, is needed. However, these findings provide useful information for hypothesis testing about potential interactions between *cidA* and *cidB*, which is exciting.

Detailed Comments

Abstract

L15- Associated may be more appropriate terminology than causes

L25 - Given that the modification-rescue terminology (Werren 1997) is in wide usage in the *Wolbachia* field (and in the main text of this paper), it is probably better to use it than the "toxin" "antitoxin" language used here. Furthermore, toxins generally refer to compounds that cause cell toxicity with antigenic properties. CI is a more specific interaction between sperm derived chromatin and the egg cytoplasm.

L25 -The language used in the abstract does not clearly indicate that the modification action of *cidB* occurs in the sperm, while the presumed rescue function of *cidA* occurs in the egg.

L26 - The abstract wording implies that domains of interaction are identified, which is not the case. However, these data provide hypotheses for what these zones could be, based on the patterns of diversity. In general, the abstract seems to claim more than the paper delivers, although the paper does deliver a lot.

L26 - It is my impression that prior work supports the sperm modification function of *cidB*, but the rescue function is hypothesized to be *cidA* acting in the egg. Thus, even the direct interactions between the two proteins is a hypothesis, not an established fact.

L27 - the emphasis on "domains of interaction" implies that the authors believe that the two molecules interact directly and physically. There is currently no data to support this, and alternatives clearly exist. Rather than biasing their findings to this particular interpretation, this model should not be emphasized to the exclusion of others, or the authors should reference data that support this direct physical interaction hypothesis. These points should also be more clearly developed in the text, including consideration of alternative models for the proposed interaction between *cidA* and *cidB*.

Main Text

L37 - The mod-resc terminology and model was presented in Werren 1997, which is not cited here. The conceptually similar (but not identical) toxin-antitoxin was first presented by Hurst 1991 (I believe). In that paper it is argued that Wolbachia could be producing a colicin-like molecule that is toxic, as well as a protector in the egg.

L37 - An interesting historical note: Other candidates have been proposed in the past, including wsp and ank.

L47 - Given that the role of cidA is debated, as you say earlier, why do you state here that it is "undoubtedly" involved?

L52 - There appears to be conflicting statements in juxtaposition here. First it is stated that there is "a demonstrated influence of host genetic background", and in the next sentence that CI diversity is "governed by wPip diversification". Some clarification is needed here, or the language needs to be modulated, for example "is also influenced by wPip diversification".

L58 - Again, there are statements next to each other that appear inconsistent. In one sentence it is stated that there is only one wPip genome sequenced, and it has one copy of the operon, but it is then concluded that "other paralogs" and other loci must be involved. What about diversity at a single operon between strains. This is not to say that an exploration of paralogs and other genes is not worthwhile; indeed it is as you have shown. But the logic flow of statements need to be tightened in the manuscript, so that apparent inconsistent statements are clarified or reworded.

L80 - Very nice addition to this study of testing different wPip genotypes in a controlled Culex genetic background.

L96 - Please briefly state how these isofemale lines with different Wolbachia were created, and provide citation if needed.

Reviewer #2 (Remarks to the Author):

Bonneau et al.

This manuscript represents a substantial advance in our understanding of the molecular basis of cytoplasmic incompatibility (CI) in *Culex pipiens* mosquitoes, long known as the best biological system for investigating this Wolbachia-based mating incompatibility. Few labs have access to the diversity of *Cx pipiens* strains used in this study. The results will be of considerable interest to those studying the molecular basis of CI.

Briefly, recent studies have shown that the *Wolbachia pipientis* *cidA-cidB* operon and/or its paralogs are strongly correlated with the modification/toxin (*CidB*) and rescue/antitoxin (*CidA*) factors associated with CI. The strength of this manuscript is the use of a collection of 183 isofemale lines of *Cx pipiens* collected worldwide to show specifically that co-variation in specific regions of the *cidA/B* genes correlates with crossing type in diverse *wPip* strains. The data are convincing, with the possible exception of copy number analyses which are difficult to assess because they could be affected by a lytic phage cycle. Regardless of whether the genes are amplified, the main finding of polymorphisms in specific regions of *CidA/B* genes is a major advance in understanding the participation of these gene products in CI. The text and figures could be reworked to describe the data more smoothly. Several comments/suggestions are listed below. In particular, a streamlined version of Extended data Figures 2 and 3 should be consolidated with Fig. 5 and made into a main figure.

Major comments:

Title: delete "amplified"

Abstract: Delete lines 20 – 22. Evidence for several copies of the operon is weak, and the mechanism is only inferred. Wouldn't the ratios be 'off' if phage were induced to some extent?

Fig. 1: Does the pattern immediately above the dotted lines above each alignment reflect the number of Illumina reads or coverage density? What do the authors make of the gaps in the *wPipIV* sequences? Could this be a technical problem related to the amplification process (Methods lines 182-185) or DNA structure (or propensity to form secondary structure) in these regions?

At the very top, what is the little red box in the *Pel* genome?

Does an alignment of the *cid* and *cin* genes show a deletion of the variable regions?

Lines 80-94: I would start with Fig 4, explain the regions of variation, blue vs. pink, then proceed to the more detailed Figs 2 and 3, which contain both blue and pink positions. In Figs. 2 and 3, the amino acid positions are hard to read, and some numbers are preceded by an unexplained "-", which makes more sense if the reader sees Fig. 4 first. In Figs. 2 and 3, amino acid positions read from top to bottom; placing them on a slant, reading left to right would possibly make it easier to read. It should be noted that the amino acids are not contiguous. A few more words could be used to explain these data. Also, fonts could be larger in the colored regions. In Fig. 4, is the whole coding sequence primarily Ankyrin-like repeats, or should there be two shades of yellow? How are Ankyrin-like repeats defined with respect to these proteins? This seems to be an inconsistency with respect to comments in lines 126-128.

Line 87: Block rearrangements are inferred, but not directly addressed by experimental data.

Lines 91-93: Extended data table 2: were the PCR primers specific for regions of the cidA/B genes that do not occur in paralogs? More detail needs to be provided in the legend. How do the authors reconcile these findings with the single operon in the sequenced Pel genome (line 59). Could the variation have anything to do with phage induction?

Line 94: All transcripts were detected: does this mean that the DNA variants were recovered at the level of RNA, cloned and sequenced? Specify how many different transcripts were detected?

Fig. 5 legend does not explain the columns with green/gray shading or the meaning of P/A and A/P. For Cid A (top panel) I don't see cidA-IVb in Figure 2; likewise, why aren't a/3 and b/3 variants shown in Fig. 3?

Line 104 only the a/2 variant was UNIQUE to incompatible lines

Line 178: I question whether centrifugation speed/time were sufficient for quantitative recovery of Wolbachia.

Extended data Fig 2: What do the dashes represent in the top line? Add size markers to the gel; explain color coding in lower panels.

Minor comments:

Lines 40 and 148: I suggest that "located in" and "belong to" be replaced with "associated with" or "located near" prophage WO modules, as I am not yet convinced that cidA/B are in fact part of the WO phage genome that excises from the Wolbachia genome. The present work does not strengthen the WO-phage connection described in reference 4.

Lines 59-61: A single operon? It's more like a complex pattern of co-evolving diversification within genes that comprise a single operon in the more general sense of the term operon; consider rewording these lines.

Line 70: list the housekeeping genes in the legend

In extended data Figure 1, spacing of numbers in boxed coordinates at left and right differs for B, vs A and C

Extended table 1, symbols have different font sizes

Line 112: specify the variants

Line 137: specify the group

Response to referees

Reviewer #1 (Remarks to the Author):

Summary: This is an excellent study exploring the diversity of the *cidA-cidB* genes in both copy number and sequence, using the model CI system *Culex pipiens*. The research is very well done, from documenting *Wolbachia* strain variation within the species, to investigating CI differences in controlled genetic backgrounds. The logic flow of some of the statements in the abstract and introduction needs improvement, apparently because the authors are anticipating their own results (e.g. copy number differences and patterns of variation in *cidA* and *cidB*).

Also, there are some missing references that should be added. I anticipate that this paper will be a significant contribution to our understanding of the genetic basis of CI variation in nature. Statements about “domains of interaction” between *cidA* and *cidB* are hypotheses. But it is described in the abstract as if it is already known that these proteins interact directly with each other, although it is not known.

More caution in interpreting these results, and consideration of alternate hypotheses, is needed. However, these findings provide useful information for hypothesis testing about potential interactions between *cidA* and *cidB*, which is exciting.

Detailed Comments

Abstract

1-L15- Associated may be more appropriate terminology than causes

Wolbachia is responsible by itself (no effect of host genetic background, of other symbionts, of host mitochondrial DNA...) for the induction of CI and for the great diversity of CI patterns in *Culex pipiens* (Duron *et al.* Mol Ecol 2006, Atyame *et al.* Mol Ecol 2011; Atyame *et al.* Plos One 2014). The use of “cause” thus seems appropriate to us.

2-L25 - Given that the modification-rescue terminology (Werren 1997) is in wide usage in the *Wolbachia* field (and in the main text of this paper), it is probably better to use it than the “toxin” “antitoxin” language used here. Furthermore, toxins generally refer to compounds that cause cell toxicity with antigenic properties. CI is a more specific interaction between sperm derived chromatin and the egg cytoplasm.

The *mod-resc* terminology does not imply any assumption on the nature of the molecules involved in the *mod* and the *resc* function. Our phenotype/genotype association results and previous publications are in favor of *cidA-cidB* operon being a classic toxin-antitoxin system, with *CidA* being the antitoxin (i.e. *resc*) of *CidB*. The arguments are: (i) the simple two-gene structure of the operon, with the putative antitoxin gene located upstream from the putative toxin gene, (ii) the overexpression of the antitoxin gene relative to the toxin gene, (iii) the ability of the two proteins to form a complex and (iv) the required co-expression of *cidA* and *cidB* for the production of live transgenic *D. melanogaster* and *Saccharomyces cerevisiae*. This is the reason why we chose to use this terminology in the abstract.

3-L25 -The language used in the abstract does not clearly indicate that the modification action of *cidB* occurs in the sperm, while the presumed rescue function of *cidA* occurs in the egg.

Functional studies by transgenesis and our phenotype/genotype data demonstrate that *CidB* is involved in the *mod* function acting in the sperm. However the case of *CidA* is more difficult to clarify. *CidA* could be the antitoxin of *CidB*, protecting *CidB* producers against its toxin harmful effects. Arguments in favor of such an hypothesis are: 1) the supposed toxicity of *CidB* towards its producer since no viable *Drosophila* expressing only *CidB* (without *CidA*) could be obtained (Beckmann *et al.* 2017) and 2) the large amount of *CidA* protein found in the sperm (in the spermathecae

of *Culex pipiens* females), which suggests that this protein might protect the sperm from the toxicity of the CidB protein it contains. It is actually this important amount that initially allowed the detection of CidA protein (Beckmann & Fallon 2013). CidA could also be involved in the *resc* function in the eggs but we do not have any available data confirming that.

4-L26 - The abstract wording implies that domains of interaction are identified, which is not the case. However, these data provide hypotheses for what these zones could be, based on the patterns of diversity. In general, the abstract seems to claim more than the paper delivers, although the paper does deliver a lot.

We agree that our data lead to the hypothesis that two zones (one in CidA and one in CidB) could be the domains of interactions between them (because they co-vary and correspond to changing in CI patterns), but do not test this hypothesis. Thus, we add the term “putative” before “domains of reciprocal interactions” (line 25).

5-L26 – It is my impression that prior work supports the sperm modification function of *cidB*, but the rescue function is hypothesized to be *cidA* acting in the egg. Thus, even the direct interactions between the two proteins is a hypothesis, not an established fact.

The fact that CidA and CidB interact with each other is actually firmly established by Beckmann *et al.* 2017 paper. In this paper pull-down experiments of co-expressed CidA/CidB proteins produced by *E coli* cells revealed that they bind directly together, which implies that they possess domains of interaction. Based on these convincing results, we do hypothesize that these proteins could interact in the variable domains we identified on both CidA and CidB. Concerning CidA acting in the egg as a *resc*, no insect functional data have been yet published to confirm that hypothesis.

6-L27 – the emphasis on “domains of interaction” implies that the authors believe that the two molecules interact directly and physically. There is currently no data to support this, and alternatives clearly exist. Rather than biasing their findings to this particular interpretation, this model should not be emphasized to the exclusion of others, or the authors should reference data that support this direct physical interaction hypothesis. These points should also be more clearly developed in the text, including consideration of alternative models for the proposed interaction between *cidA* and *cidB*.

As said before, and after consultation of several biochemists, the biochemical data obtained by Beckmann *et al.* (2017) do demonstrate that CidA and CidB directly and physically interact with each other. We do agree with the referee that we did not explain enough in the previous version of the paper what the actual proofs of direct interaction between CidA and CidB are, and thus added a sentence about pull-down experiments (line 39-40).

Main Text

7-L37 - The mod-*resc* terminology and model was presented in Werren 1997, which is not cited here. The conceptually similar (but not identical) toxin-antitoxin was first presented by Hurst 1991 (I believe). In that paper it is argued that *Wolbachia* could be producing a colicin-like molecule that is toxic, as well as a protector in the egg.

It is true that in the final version of the paper we forgot to cite the seminal paper of Werren 1997. In our paper, we focused on the toxin-antitoxin model as CidA-CidB exhibit many genomic and genetic features of a toxin-antitoxin system. We now cite the Werren paper as it should be (line 32).

8-L37 – An interesting historical note: Other candidates have been proposed in the past, including *wsp* and *ank*.

It is true that other genes have been proposed in the past, especially the genes containing ankyrin repeats due to their role in (i) protein-protein interactions, (ii) cellular cycle regulation and (iii) gene regulation. As the CidA-CidB proteins putatively contain many Ankyrin repeat domains, the hypothesis that *ank-ank* interactions could be fundamental in CI mechanisms is even fostered by our data. We changed the text according to those lines 32-37 but also lines 149-156

9-L47 – Given that the role of *cidA* is debated, as you say earlier, why do you state here that it is “undoubtedly” involved?

It is the exact role of CidA that is debated: is it the resc? Is it an elicitor of CidB? Is it just the antitoxin of CidB without being the resc? However, its implication in CI has been highly suggested since in Lepage *et al.* 2017 the authors showed that both *cifA* and *cifB* are required to induce CI and since Beckman *et al.* 2017 showed that CidA prevented the toxicity of CidB when co-expressed in yeast or in *Drosophila*. These two papers proposed quite opposite roles to CidA, but both concluded to its preponderant role in CI. However, we do agree with the referee that the term “undoubtedly” is too strong; we thus get rid of it (line 49).

10-L52 – There appears to be conflicting statements in juxtaposition here. First it is stated that there is “a demonstrated influence of host genetic background”, and in the next sentence that CI diversity is “governed by wPip diversification”. Some clarification is needed here, or the language needs to be modulated, for example “ is also influenced by wPip diversification”.

The language used indeed made the sentence equivocal. In the *Culex pipiens* mosquitoes, all our previous studies demonstrated that the CI patterns were driven by the diversity of wPip strains, and no effect of the host genetic background was ever detected. We changed the text to clarify this sentence (line 53-55).

11-L58 – Again, there are statements next to each other that appear inconsistent. In one sentence it is stated that there is only one wPip genome sequenced, and it has one copy of the operon, but it is then concluded that “other paralogs” and other loci must be involved. What about diversity at a single operon between strains. This is not to say that an exploration of paralogs and other genes is not worthwhile; indeed it is as you have shown. But the logic flow of statements need to be tightened in the manuscript, so that apparent inconsistent statements are clarified or reworded.

We do understand that statements could appear inconsistent, but it was a misunderstanding that we hopefully clarified in the present version of the manuscript. What was inferred from crossing data before the present paper was that several “genes” coding for different *mod* and *resc* had to co-exist within the same wPip genome to explain CI properties of a single *C. pipiens* isofemale line (see Atyame *et al.* 2011 and Nor *et al.* 2013). This is the reason why we said that i) totally different genes, or (ii) paralogs of *cidA-cidB* (e.g. *cinA-cinB*), or (iii) multiple variable copies of *cidA-cidB* operon (within the same genome) are required to explain CI diversity in wPip. In the revised version, we clearly call “copies” or “variants” the different versions of the *cidA-cidB* genes that were found in wPip genomes, and we call paralogs the divergent genes like *cinA* and *cinB*. We hope the rewriting of the text from line 58 to line 64 will improve the comprehension of future readers.

12-L96 – Please briefly state how these isofemale lines with different *Wolbachia* were created, and provide citation if needed.

The referee is right, the procedure was not detailed enough. We thus changed the main text line 100-101: “Each of these isofemale lines was founded with one initial egg-raft from a single female.”, and Material and Method line 217-220: “*C. pipiens* larvae and pupae were collected in the field and reared to adulthood in the laboratory. Females were then fed on blood to lay eggs that served to establish isofemale lines. Each egg raft (containing 100–300 eggs) was individually isolated for hatching, and the *Wolbachia* group present was determined by performing *pk1* PCR-RFLP tests on two first-instar larvae (L1). Isofemale lines were created by rearing the offspring resulting from a single egg-raft (thus from a single female)”

Reviewer #2 (Remarks to the Author):

Bonneau et al.

This manuscript represents a substantial advance in our understanding of the molecular basis of cytoplasmic incompatibility (CI) in *Culex pipiens* mosquitoes, long known as the best biological system for investigating this *Wolbachia*-based mating incompatibility. Few labs have access to the diversity of *Cx pipiens* strains used in this study. The results will be of considerable interest to those studying the molecular basis of CI.

Briefly, recent studies have shown that the *Wolbachia pipiens* *cidA-cidB* operon and/or its paralogs are strongly correlated with the modification/toxin (CidB) and rescue/antitoxin (CidA) factors associated with CI. The strength of this manuscript is the use of a collection of 183 isofemale lines of *Cx pipiens* collected worldwide to show specifically

that co-variation in specific regions of the *cidA/B* genes correlates with crossing type in diverse *wPip* strains. The data are convincing, with the possible exception of copy number analyses which are difficult to assess because they could be affected by a lytic phage cycle. Regardless of whether the genes are amplified, the main finding of polymorphisms in specific regions of *CidA/B* genes is a major advance in understanding the participation of these gene products in CI. The text and figures could be reworked to describe the data more smoothly. Several comments/suggestions are listed below. In particular, a streamlined version of Extended data Figures 2 and 3 should be consolidated with Fig. 5 and made into a main figure.

Major comments:

1-Title: delete “amplified”

We do understand that the referee doubts our utilization of the term “amplification” because of the impact that lytic phage cycles could have on q-PCR quantification of the number of *cidA* and *cidB* copies per *wPip* genome. However, we do have strong arguments that plead in favor of this genic amplification:

- (i) The first argument is the diversity of the *cidA* and *cidB* copies we uncovered within each host. Such diversity in a single individual cannot be explained by phage lysis amplification.
- (ii) The second argument is the stability of this diversity. We indeed sequenced Istanbul individuals conserved in liquid nitrogen from 2006, 2012 and 2017. We found the same exact copies (variants) of *cidA* and *cidB* genes in all these individuals. These data exclude the hypothesis of a co-infection with different *wPip* strains within the same host harbouring distinct *cidA* and *cidB* genes. This new data will hopefully convince the referee.
- (iii) A last argument is that the same variant repertoire of *cidA* and *cidB* were found in the group *wPipIV* in *wPip* strains as geographically distant as China, Tunisia, Algeria and Turkey.

We modified the text line 163-170 to more clearly explain our arguments. Concerning q-PCR, we agree with the referee that we cannot exclude the possibility of an overestimation of the copy number due to lytic phages. Previous studies have demonstrated that phage multiplication can be responsible for apparent gene amplification without actual amplification of the genes in the genome (Bordenstein *et al.* 2006). The present version explains that we have more robust arguments for amplification based on the stability of the *cidA* and *cidB* repertoire both in time and in space.

2-Abstract: Delete lines 20 – 22. Evidence for several copies of the operon is weak, and the mechanism is only inferred. Wouldn't the ratios be 'off' if phage were induced to some extent?

For the reason we listed above, we truly believe that we have convincing arguments that *cidA* and *cidB* are present in several copies in each single *wPip* genome.

3-Fig. 1: Does the pattern immediately above the dotted lines above each alignment reflect the number of Illumina reads or coverage density?

The pattern immediately above the dotted line reflects the number of Illumina reads that have mapped for each position. This is now stated in the legend of the figure.

4-What do the authors make of the gaps in the *wPipIV* sequences? Could this be a technical problem related to the amplification process (Methods lines 182-185) or DNA structure (or propensity to form secondary structure) in these regions?

The main explanation for the gaps observed in the mapping of the *wPip* reads from Tunis, Harash and Istanbul on the reference genome *wPip_Pel* is the differing phylogenetic distances between the *wPip* infecting these three lines and the reference genome of *wPip_Pel*. Only one copy of *cidA* and *cidB* is present in *wPip_Pel* genome; with the mapping program and parameters we used, the reads harbouring sequences that are too distant from the one in the *wPip_Pel* assembled genome did not map. The *Wolbachia* strain *wPip_Pel* belongs to the *wPip-I* group as the *wPip* from Tunis: the referee can check that there are fewer gaps on the mapping of the “Tunis reads” onto *wPip_Pel* than with Harash

and Istanbul reads, which belong to the *wPipIV* group. Moreover, the location of the “mapping gaps” is not random and is quite coherent between *wPipIV* group Harash and Istanbul, thus showing that they are closer to each other.

We believe that *cidA* and *cidB* are also amplified in the *wPip_Pel* genome: in each *Culex pipiens* individual we sequenced so far (at least 50), all presented several sequences of *cidA* and *cidB*. The fact that only a single operon is reported in the assembled and published genome is probably an assembly error that contributes to the confusion.

5-At the very top, what is the little red box in the Pel genome?

The little red box in the *wPel* genome allows the IGV user to locate, in the *wPip_Pel* reference genome, the zone that is visualized with more details on the bottom panels. In A, it indicates the location of *cidA* and *cidB* in the *wPel* genome. We changed the legend of the figure 1 in accordance with the referee's remark.

6-Does an alignment of the *cid* and *cin* genes show a deletion of the variable regions?

cidA variants and *cinA* sequence (the same for *cidB* variants and *cinB* sequence) are too distant to be aligned correctly. The identity between *cidA* and *cinA* is around 30% and around 20% for *cidB* and *cinB* (see Beckman and Fallon 2013). This is thus impossible to answer the referee's question.

7-Lines 80-94: I would start with Fig 4, explain the regions of variation, blue vs. pink, then proceed to the more detailed Figs 2 and 3, which contain both blue and pink positions.

We realized that we used blue and pink colors for two different things in different figures, which is for sure confusing for the reader.

-In new Figure 3: Blue and pink colors represent the two gene portions (same position but different AA sequences) that splits the network phylogeny of CidA variants (found in all *wPip* groups) into two subgroups.

-In new Figure 6: Blue and pink colors represent the variable zones in both genes within the *wPip IV* group. There is no link between the blue and the pink between the new figure 3 (and supplementary figure 2) and previous figure 4. We thus changed the colors in the new Figure 6 to avoid the confusion.

We cannot start with previous Figure 4, because the aim of this figure is to visualize the putative zones of interaction between CidA and CidB, due to the variations observed in *wPip-IV*. However, we decided, as suggested by the reviewer, to make new figures (new Figure 3,4 and 6) composed of a new panel A, which is based on the global representation of the protein and its variation used in the former Figure 4.

8-In Figs. 2 and 3, the amino acid positions are hard to read, and some numbers are preceded by an unexplained “-“, which makes more sense if the reader sees Fig. 4 first

Explanation for the “-“ was added in Figure 3 and 4 legends.

9-In Figs. 2 and 3, amino acid positions read from top to bottom; placing them on a slant, reading left to right would possibly make it easier to read.

We tried to modify the table according to the referee suggestions, but it did not improved its readability. However, we changed the font size, hoping it will make it easier to read.

10-It should be noted that the amino acids are not contiguous.

The legend of the new figures 3 and 4 were changed accordingly.

11-A few more words could be used to explain these data.

The legend of the new figures were changed accordingly

12-Also, fonts could be larger in the colored regions.

Font size of amino acid and position were enlarged in new Figure 3 and 4, as suggested by the referee.

13- In Fig. 4, is the whole coding sequence primarily Ankyrin-like repeats, or should there be two shades of yellow? How are Ankyrin-like repeats defined with respect to these proteins? These seems to be an inconsistency with respect to comments in lines 126-128.

We agree with the reviewer and clarified this part line.

We extended the explanation within the text (line 146-156) and changed the figure legend to carefully explain our results and hypothesis.

14-Line 87: Block rearrangements are inferred, but not directly addressed by experimental data. We changed “demonstrated” by “suggested” (line 106-108).

15-Lines 91-93: Extended data table 2: were the PCR primers specific for regions of the *cidA/B* genes that do not occur in paralogs?

The primers for *cidA* and *cidB* do not match on the sequence of *cinA* and *cinB*. The polymorphism we revealed is not resulting from a mixed amplification of *cinA* and *cinB* together with *cidA* and *cidB*. Moreover, this is also coherent with Illumina read analyses.

16-More detail needs to be provided in the legend (extended data table 2). How do the authors reconcile these findings with the single operon in the sequenced Pel genome (line 59). Could the variation have anything to do with phage induction?

As previously said (reviewer 2, comment 4), the presence of only one copy of the *cidA/cidB* operon in the reference genome *wPip_Pel* is most likely an assembly error. We know how hard it is to assemble a *Wolbachia* genome with so many repeats. We changed the discussion to present better arguments about *cidA* and *cidB* being amplified and say clearly that *wPip_Pel* is certainly misassembled in this region (line 161-172).

17-Line 94: All transcripts were detected: does this mean that the DNA variants were recovered at the level of RNA, cloned and sequenced? Specify how many different transcripts were detected?

We extracted total RNAs from mosquitoes from the Istanbul line and reverse transcribed them (using random primers). Then, we cloned and sequenced the *cidA* and *cidB* cDNA using the same primers as the one used for the PCR on DNA (Figure extended data 2 and 3). The six *cidA* variants and the four *cidB* variants were all found in sequences from cDNA. This results means that all the variants of *cidA* and *cidB* present in the genome of the *wPip* strain infecting Istanbul line are expressed. We changed the text line 114-117 to better explain this.

18-Fig. 5 legend does not explain the columns with green/gray shading or the meaning of P/A and A/P.

We changed the figure legend to explain green and gray and the meaning of P/A and A/P.

19-For Cid A (top panel) I don't see *cidA-IVb* in Figure 2; likewise, why aren't a/3 and b/3 variants shown in Fig. 3?

In Figure 3 only the variants of the *wPip* strain (group IV) from Istanbul line are represented. *wPip* Istanbul does not harbour *cidA_IV(β)*, *cidB_IV(a/3)* and *cidB_IV(b3)*. This is the reason why they are not mentioned in Figure 3.

20-Line 104 only the a/2 variant was UNIQUE to incompatible lines

cidB_IV(a/2) is indeed specific to incompatible lines but we also found *cidA_IV(δ)* to be also specific to these incompatible lines. Thus we modified the main text to clarify this point (line 130-131).

21-Line 178: I question whether centrifugation speed/time were sufficient for quantitative recovery of *Wolbachia*.

This protocol was inspired from Ellegaard *et al.* 2013 and worked for us with these centrifugation speed and time. However, we misconverted the rpm to the g: the speed was actually 6,000g and not 5,400g. We changed the methods text accordingly (line 240-241).

22-Extended data Fig 2: What do the dashes represent in the top line?

The black line represents the *cidA_IV* sequence with a focus on the polymorphic region. Non-polymorphic regions were shortened by representing them with dashed lines. The figure legend was changed accordingly.

23-Add size markers to the gel; explain color coding in lower panels.

Modifications to figure and legend were done.

Minor comments:

24-Lines 40 and 148: I suggest that “located in” and “belong to” be replaced with “associated with” or “located near” prophage WO modules, as I am not yet convinced that *cidA/B* are in fact part of the WO phage genome that excises from the *Wolbachia* genome. The present work does not strengthen the WO-phage connection described in reference 4.

It is true that *cidA/cidB* operon is in the phage region but might not be part of the phage and also not part of the lytic version of the phage. We changed “belong to” into “associated with” (line 40-42).

25-Lines 59-61: A single operon? It’s more like a complex pattern of co-evolving diversification within genes that comprise a single operon in the more general sense of the term operon; consider rewording these lines.

This part was deeply rewritten, taking into account remarks of referee 1 that also suggested to clarify it (line 58-67).

26-Line 70: list the housekeeping genes in the legend

Done 27-In extended data Figure 1, spacing of numbers in boxed coordinates at left and right differs for B, vs A and C
modified

28-Extended table 1, symbols have different font sizes
modified

29-Line 112: specify the variants

Done

30-Line 137: specify the group

Done

REVIEWERS' COMMENTS:

Reviewer #1 (Remarks to the Author):

The clarity of the paper has been improved, as well as the discussion of models for CI. This paper will make an important contribution to our understanding of CI mechanisms. I look forward to further advances as studies are conducted on biochemical interactions among variants in the mod (toxin) rescue (antitoxin) system.

Reviewer #2 (Remarks to the Author):

This manuscript has been nicely revised, and satisfies prior reviewer comments. I have a few minor comments:

line 34: latter (delete S)

Line 65 : consider change to "cinA-cinB paralog with predicted nuclease activity."

line 74 consider changing "the reciprocal" to "physical"

line 173, should the word "only" be deleted? There may be other systems we don't know about.

Response to referees

REVIEWERS' COMMENTS:

Reviewer #1 (Remarks to the Author):

The clarity of the paper has been improved, as well as the discussion of models for CI. This paper will make an important contribution to our understanding of CI mechanisms. I look forward to further advances as studies are conducted on biochemical interactions among variants in the mod (toxin) rescue (antitoxin) system.

Reviewer #2 (Remarks to the Author):

This manuscript has been nicely revised, and satisfies prior reviewer comments. I have a few minor comments:

line 34: latter (delete S)

Done

Line 65 : consider change to "cinA-cinB paralog with predicted nuclease activity."

Done

line 74 consider changing "the reciprocal" to "physical"

Done

line 173, should the word "only" be deleted? There may be other systems we don't know about

Done